# Connecting Certified and Adversarial Training

**Yuhao Mao, Mark Niklas Müller, Marc Fischer, Martin Vechev**
Department of Computer Science
ETH Zurich, Switzerland
{yuhao.mao, mark.mueller, marc.fischer, martin.vechev}@inf.ethz.ch

## Abstract

Training certifiably robust neural networks remains a notoriously hard problem. While adversarial training optimizes *under-approximations* of the worst-case loss, which leads to insufficient regularization for certification, sound certified training methods, optimize loose *over-approximations*, leading to over-regularization and poor (standard) accuracy. In this work, we propose TAPS, an (unsound) certified training method that combines IBP and PGD training to optimize more precise, although not necessarily sound, worst-case loss approximations, reducing over-regularization and increasing certified and standard accuracies. Empirically, TAPS achieves a new state-of-the-art in many settings, e.g., reaching a certified accuracy of 22% on TINYIMAGENET for $\ell_\infty$-perturbations with radius $\epsilon = 1/255$. We make our implementation and networks public at github.com/eth-sri/taps.

## 1   Introduction

Adversarial robustness, *i.e.*, a neural network's resilience to small input perturbations (Biggio et al., 2013; Szegedy et al., 2014), has established itself as an important research area.

**Neural Network Certification**   can rigorously prove such robustness: While complete verification methods (Tjeng et al., 2019; Bunel et al., 2020; Zhang et al., 2022a; Ferrari et al., 2022) can decide every robustness property given enough (exponential) time, incomplete methods (Wong and Kolter, 2018; Singh et al., 2019a; Zhang et al., 2018) trade precision for scalability.

**Adversarial training**   methods, such as PGD (Madry et al., 2018), aim to improve robustness by training with samples that are perturbed to approximately maximize the training loss. This can be seen as optimizing an *under-approximation* of the worst-case loss. While it *empirically* improves robustness significantly, it generally does not induce sufficient regularization for certification and has been shown to fail in the face of more powerful attacks (Tramèr et al., 2020).

**Certified Training**   methods, in contrast, optimize approximations of the worst-case loss, thus increasing certified accuracies at the cost of over-regularization that leads to reduced standard accuracies. In this work, we distinguish two certified training paradigms. Sound methods (Mirman et al., 2018; Gowal et al., 2018; Shi et al., 2021) compute sound over-approximations of the worst-case loss via bound propagation. The resulting approximation errors induce a strong (over-)regularization that makes certification easy but causes severely reduced standard accuracies. Interestingly, reducing these approximation errors by using more precise bound propagation methods, empirically, results in strictly *worse* performance, as they induce harder optimization problems (Jovanović et al., 2022). This gave rise to unsound methods (Balunovic and Vechev, 2020; Palma et al., 2022; Müller et al., 2022a), which aim to compute *precise* but not necessarily sound approximations of the worst-case loss, reducing (over-)regularization and resulting in networks that achieve higher standard and certified accuracies, but can be harder to certify. Recent advances in certification techniques, however, have made their certification practically feasible (Ferrari et al., 2022; Zhang et al., 2022a).

37th Conference on Neural Information Processing Systems (NeurIPS 2023).

We illustrate this in Figure 1, where we compare certified training methods with regard to their worst-case loss approximation errors and the resulting trade-off between certified and standard accuracy. On the left, we show histograms of the worst-case loss approximation error over test set samples (see Section 4.2 for more details). Positive values (right of the y-axis) correspond to over- and negative values (left of the y-axis) to under-approximations. As expected, we observe that the sound IBP (Gowal et al., 2018) always yields over-approximations (positive values) while the unsound SABR (Müller et al., 2022a) yields a more precise (6-fold reduction in mean error) but unsound approximation of the worst-case loss. Comparing the resulting accuracies (right), we observe that this more precise approximation of the actual optimization objective, *i.e.*, the true worst-case loss, by SABR (◆) yields both higher certified and standard accuracies than the over-approximation by IBP (●). Intuitively, reducing the over-regularization induced by a systematic underestimation of the network's robustness allows it to allocate more capacity to making accurate predictions.

The core challenge of effective certified training is, thus, to compute precise (small mean error and low variance) worst-case loss approximations that induce a well-behaved optimization problem.

**This Work** proposes **T**raining via **A**dversarial **P**ropagation through **S**ubnetworks (TAPS), a novel (unsound) certified training method tackling this challenge, thereby increasing both certified and standard accuracies. Compared to SABR (◆ the current state-of-the-art), TAPS (■) enjoys a further 5-fold mean approximation error reduction and significantly reduced variance (Figure 1 left), leading to improved certified and natural accuracies (right). The key technical insight behind TAPS is to combine IBP and PGD training via a gradient connector, a novel mechanism that allows training the whole network jointly such that the over-approximation of IBP

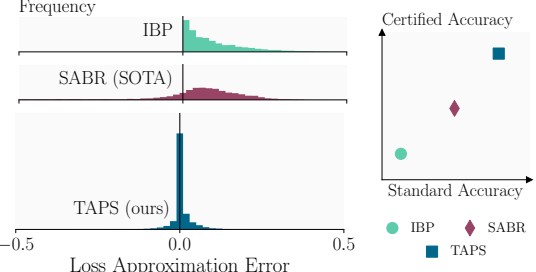

Figure 1: Histograms of the worst-case loss approximation errors over the test set (left) for different training methods show that TAPS (our work) achieves the most precise approximations and highest certified accuracy (right). Results shown here are for a small CNN3.

and under-approximations of PGD cancel out. We demonstrate in an extensive empirical study that TAPS yields exceptionally tight worst-case loss approximations which allow it to improve on state-of-the-art results for MNIST, CIFAR-10, and TINYIMAGENET.

## 2 Background on Adversarial and Certified Training

Here, we provide the necessary background on adversarial and certified training. We consider a classifier $F: \mathcal{X} \mapsto \mathcal{Y}$ parameterized by weights $\boldsymbol{\theta}$ and predicting a class $y_{\text{pred}} \coloneqq F(\boldsymbol{x}) \coloneqq \arg\max_{y \in \mathcal{Y}} f_y(x)$ for every input $\boldsymbol{x} \in \mathcal{X} \subseteq \mathbb{R}^d$ with label $y \in \mathcal{Y} \coloneqq \{1, \ldots, K\}$ where $\boldsymbol{f}: \mathcal{X} \mapsto \mathbb{R}^{|\mathcal{Y}|}$ is a neural network, assigning a numerical logit $o_i \coloneqq f_i(\boldsymbol{x})$ to each class $i$.

**Adversarial Robustness** We call a classifier adversarially robust on an $\ell_p$-norm ball $\mathcal{B}_p(\boldsymbol{x}, \epsilon)$ if it classifies all elements within the ball to the correct class, *i.e.*, $F(\boldsymbol{x}') = y$ for all perturbed inputs $\boldsymbol{x}' \in \mathcal{B}_p(\boldsymbol{x}, \epsilon)$. In this work, we focus on $\ell_\infty$-robustness with $\mathcal{B}_\infty(\boldsymbol{x}, \epsilon) \coloneqq \{\boldsymbol{x}' \mid \|\boldsymbol{x}' - \boldsymbol{x}\|_\infty \leq \epsilon\}$ and thus drop the subscript $\infty$.

**Neural Network Certification** is used to formally *prove* robustness properties of a neural network, *i.e.*, that all inputs in the region $\mathcal{B}(\boldsymbol{x}, \epsilon)$ yield the correct classification. We call samples $\boldsymbol{x}$ where this is successfull, certifiably robust and denote the portion of such samples as *certified accuracy*, forming a lower bound to the true robustness of the analyzed network.

Interval bound propagation (IBP) (Mirman et al., 2018; Gowal et al., 2018) is a particularly simple yet effective certification method. Conceptually, it computes an over-approximation of a network's reachable set by propagating the input region $\mathcal{B}(\boldsymbol{x}, \epsilon)$ through the network, before checking whether all reachable outputs yield the correct classification. This is done by, first, over-approximating the input region $\mathcal{B}(\boldsymbol{x}, \epsilon)$ as a BOX $[\underline{\boldsymbol{x}}^0, \overline{\boldsymbol{x}}^0]$ (each dimension is described as an interval), centered at $\boldsymbol{c}^0 = \boldsymbol{x}$ and with radius $\boldsymbol{\delta}^0 = \epsilon$, such that we have the $i$th dimension of the input $x_i^0 \in [c_i^0 - \delta_i^0, c_i^0 + \delta_i^0]$. We then

propagate it through the network layer-by-layer (for more details, see (Mirman et al., 2018; Gowal et al., 2018)), until we obtain upper and lower bounds $[\underline{o}^\Delta, \overline{o}^\Delta]$ on the logit differences $o^\Delta := o - o_y \mathbf{1}$. If we can now show dimensionwise that $\overline{o}^\Delta < 0$ (except for $\overline{o}_y^\Delta = 0$), this proves robustness. Note that this is equivalent to showing that the maximum margin loss $\mathcal{L}_{\text{MA}}(x', y) := \max_{i \neq y} \overline{o}_i^\Delta$ is less than 0 for all perturbed inputs $x' \in \mathcal{B}(x, \epsilon)$.

**Training for Robustness** aims to find a model parametrization $\theta$ that minimizes the expected worst-case loss for some loss-function $\mathcal{L}$:

$$\theta = \arg\min_{\theta} \mathbb{E}_{x,y} \left[ \max_{x' \in \mathcal{B}(x,\epsilon)} \mathcal{L}(x', y) \right]. \tag{1}$$

As the inner maximization objective in Equation (1) can generally not be solved exactly, it is often under- or over-approximated, giving rise to adversarial and certified training, respectively.

**Adversarial Training** optimizes a lower bound on the inner maximization problem in Equation (1) by training the network with concrete samples $x' \in \mathcal{B}(x, \epsilon)$ that (approximately) maximize the loss function. A well-established method for this is *Projected Gradient Descent (PGD)* training (Madry et al., 2018) which uses the Cross-Entropy loss $\mathcal{L}_{\text{CE}}(x, y) := \ln\left(1 + \sum_{i \neq y} \exp(f_i(x) - f_y(x))\right)$. Starting from a random initialization point $\hat{x}_0 \in \mathcal{B}(x, \epsilon)$, it performs $N$ update steps

$$\hat{x}_{n+1} = \Pi_{\mathcal{B}(x,\epsilon)} \hat{x}_n + \eta \, \text{sign}(\nabla_{\hat{x}_n} \mathcal{L}(\hat{x}_n, y))$$

with step size $\eta$ and projection operator $\Pi$. Networks trained this way typically exhibit good empirical robustness but remain hard to formally certify and vulnerable to stronger or different attacks (Tramèr et al., 2020; Croce and Hein, 2020).

**Certified Training**, in contrast, is used to train *certifiably* robust networks. In this work, we distinguish two classes of such methods: while *sound* methods optimize a sound upper bound of the inner maximization objective in Equation (1), *unsound* methods sacrifice soundness to use an (in expectation) more precise approximation. Methods in both paradigms are often based on evaluating the cross-entropy loss $\mathcal{L}_{\text{CE}}$ with upper bounds on the logit differences $\overline{o}^\Delta$.

IBP (a sound method) uses sound BOX bounds on the logit differences, yielding

$$\mathcal{L}_{\text{IBP}}(x, y, \epsilon) := \ln\left(1 + \sum_{i \neq y} \exp(\overline{o}_i^\Delta)\right). \tag{2}$$

SABR (an unsound method) (Müller et al., 2022a), in contrast, first searches for an adversarial example $x' \in \mathcal{B}(x', \epsilon - \tau)$ and then computes BOX-bounds only for a small region $\mathcal{B}(x', \tau) \subset \mathcal{B}(x, \epsilon)$ (with $\tau < \epsilon$) around this adversarial example $x'$ instead of the original input $x$

$$\mathcal{L}_{\text{SABR}} := \max_{x' \in \mathcal{B}(x', \epsilon - \tau)} \mathcal{L}_{\text{IBP}}(x', y, \tau). \tag{3}$$

This generally yields a more precise (although not sound) worst-case loss approximation, thereby reducing over-regularization and improving both standard and certified accuracy.

## 3 Precise Worst-Case Loss Approximation

In this section, we first introduce TAPS, a novel certified training method combining IBP and PGD training to obtain more precise worst-case loss estimates, before showing that this approach is orthogonal and complementary to current state-of-the-art methods.

### 3.1 TAPS – Combining IBP and PGD

The key insight behind TAPS is that adversarial training with PGD and certified training with IBP complement each other perfectly: (i) both yield well-behaved optimization problems, as witnessed by their empirical success, and (ii) we can combine them such that the over-approximation errors incurred during IBP are compensated by the under-approximations of PGD. TAPS harnesses this as follows: For every sample, we first propagate the input region part-way through the network using

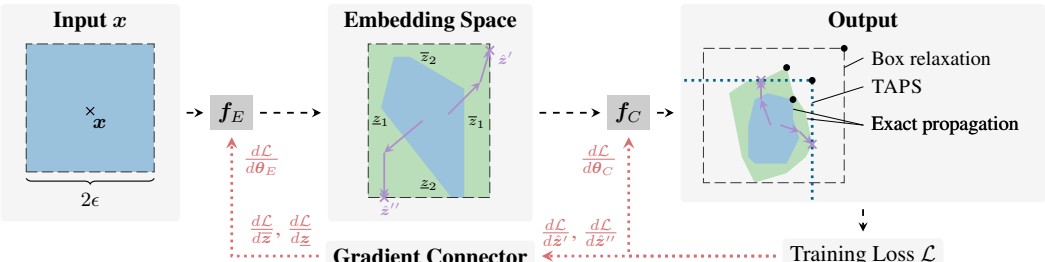

Figure 2: Overview of TAPS training. First, forward propagation (-→) of a region $B(\boldsymbol{x}, \epsilon)$ (■, left) around an input $\boldsymbol{x}$ (×) through the feature extractor $\boldsymbol{f}_E$ yields the exact reachable set (■, middle) and its IBP approximation $[\underline{\boldsymbol{z}}, \overline{\boldsymbol{z}}]$ (⬚, middle) in the embedding space. Further IBP propagation through the classifier $\boldsymbol{f}_C$ would yield an imprecise box approximation (⬚, right) of the reachable set (■, right). Instead, TAPS conducts an adversarial attack (→) in the embedding space IBP approximation (●) yielding an under-approximation (⬚) of its reachable set (■, right). We illustrate the points realizing the worst-case loss in every output region with • and enable back-propagation (·→) through the adversarial attack by introducing the gradient connector (discussed in Section 3.2).

IBP and then conduct PGD training within the thus obtained Box approximation. The key technical challenge with this approach lies in connecting the gradients of the two propagation methods and thus enabling joint training of the corresponding network portions.

We now explain TAPS in more detail along the illustration in Figure 2. We first partition a neural network $\boldsymbol{f}$ with weights $\boldsymbol{\theta}$ into a *feature extractor* $\boldsymbol{f}_E$ and a *classifier* $\boldsymbol{f}_C$ with parameters $\boldsymbol{\theta}_E$ and $\boldsymbol{\theta}_C$, respectively, such that we have $\boldsymbol{f}_{\boldsymbol{\theta}} = \boldsymbol{f}_C \circ \boldsymbol{f}_E$ and $\boldsymbol{\theta} = \boldsymbol{\theta}_E \cup \boldsymbol{\theta}_C$. We refer to the output space of the feature extractor as the *embedding space*. Given an input sample $\boldsymbol{x}$ (illustrated as × in Figure 2) and a corresponding input region $\mathcal{B}(\boldsymbol{x}, \epsilon)$ (■ in the input panel), training proceeds as follows: During the forward pass (black dashed arrows -→), we first use IBP to compute a Box over-approximation $[\underline{\boldsymbol{z}}, \overline{\boldsymbol{z}}]$ (dashed box ⬚) of the feature extractor's exact reachable set (blue region ■), shown in the middle panel of Figure 2. Then, we conduct separate adversarial attacks (→) within this region in the embedding space (●) to bound all output dimensions of the classifier. This yields latent adversarial examples $\hat{\boldsymbol{z}} \in [\underline{\boldsymbol{z}}, \overline{\boldsymbol{z}}]$, defining the TAPS bounds $\overline{\boldsymbol{o}}_{\mathrm{TAPS}}^{\triangle}$ (dotted lines ⬚ in the output space) on the network's output. This way, the *under-approximation* of the classifier via PGD, partially compensates the *over-approximation* of the feature extractor via IBP. Full IBP propagation, in contrast, continues to exponentially accumulate approximation errors (Müller et al., 2022a; Shi et al., 2021), yielding the much larger dashed box ⬚. We now compute the TAPS loss $\mathcal{L}_{\mathrm{TAPS}}$ analogously to $\mathcal{L}_{\mathrm{IBP}}$ (Equation (2)) by plugging the TAPS bound estimate $\overline{\boldsymbol{o}}_{\mathrm{TAPS}}^{\triangle}$ into the Cross-Entropy loss. Comparing the resulting losses (illustrated as • and growing towards the top right), we see that while the TAPS bounds are not necessarily sound, they yield a much better approximation of the true worst-case loss.

During the backward pass (orange dotted arrows ·→ in Figure 2), we compute the gradients w.r.t. the classifier's parameters $\boldsymbol{\theta}_C$ and the latent adversarial examples $\hat{\boldsymbol{z}}$ (classifier input) as usual. However, to compute the gradients w.r.t. the feature extractor's parameters $\boldsymbol{\theta}_F$, we have to compute (pseudo) gradients of the latent adversarial examples $\hat{\boldsymbol{z}}$ w.r.t. the box bounds $\underline{\boldsymbol{z}}$ and $\overline{\boldsymbol{z}}$. As these gradients are not well defined, we introduce the *gradient connector*, discussed next, as an interface between the feature extractor and classifier, imposing such pseudo gradients. This allows us to train $\boldsymbol{f}_E$ and $\boldsymbol{f}_C$ *jointly*, leading to a feature extractor that minimizes approximation errors and a classifier that is resilient to the spurious points included in the remaining approximation errors.

## 3.2 Gradient Connector

The key function of the gradient connector is to enable gradient computation through the adversarial example search in the embedding space. Using the chain rule, this only requires us to define the (pseudo) gradients $\frac{\partial \hat{\boldsymbol{z}}}{\partial \underline{\boldsymbol{z}}}$ and $\frac{\partial \hat{\boldsymbol{z}}}{\partial \overline{\boldsymbol{z}}}$ of the latent adversarial examples $\hat{\boldsymbol{z}}$ w.r.t. the box bounds $\underline{\boldsymbol{z}}$ and $\overline{\boldsymbol{z}}$ on the feature extractor's outputs. Below, we will focus on the $i^{\mathrm{th}}$ dimension of the lower box bound $\underline{z}_i$ and note that all other dimensions and the upper bounds follow analogously.

As the latent adversarial examples can be seen as multivariate functions in the box bounds, we obtain the general form $\frac{d\mathcal{L}}{d\underline{z}_i} = \sum_j \frac{d\mathcal{L}}{d\hat{z}_j} \frac{\partial \hat{z}_j}{\partial \underline{z}_i}$, depending on all dimensions of the latent adversarial example.

We now consider a single PGD step and observe that bounds in the $i^{\text{th}}$ dimension have no impact on the $j^{\text{th}}$ coordinate of the resulting adversarial example as they impact neither the gradient sign nor the projection in this dimension, as BOX bounds are axis parallel. We thus assume independence of the $j^{\text{th}}$ dimension of the latent adversarial example $\hat{z}_j$ from the bounds in the $i^{\text{th}}$ dimension $\underline{z}_i$ and $\overline{z}_i$ (for $i \neq j$), which holds rigorously (up to initialization) for a single step attack and constitutes a mild assumption for multi-step attacks. Therefore, we have $\frac{\partial \hat{z}_j}{\partial \underline{z}_i} = 0$ for $i \neq j$ and obtain $\frac{d\mathcal{L}}{d\underline{z}_i} = \frac{d\mathcal{L}}{d\hat{z}_i} \frac{\partial \hat{z}_i}{\partial \underline{z}_i}$, leaving only $\frac{\partial \hat{z}_i}{\partial \underline{z}_i}$ for us to define.

The most natural gradient connector is the *binary connector*, *i.e.*, set $\frac{\partial \hat{z}_i}{\partial \underline{z}_i} = 1$ when $\hat{z}_i = \underline{z}_i$ and $0$ otherwise, as it is a valid sub-gradient for the projection operation in PGD. However, the latent adversarial input often does not lie on a corner (extremal vertex) of the BOX approximation, leading to sparse gradients and thus a less well-behaved optimization problem. More importantly, the binary connector is very sensitive to the distance between (local) loss extrema and the box boundary and thus inherently ill-suited to gradient-based optimization. For example, a local extremum at $\hat{z}_i$ would induce $\frac{\partial \hat{z}_i}{\partial \underline{z}_i} = 1$ in the box $[\hat{z}_i, 0]$, but $\frac{\partial \hat{z}_i}{\partial \underline{z}_i} = 0$ for $[\hat{z}_i - \epsilon, 0]$, even for arbitrarily small $\epsilon$.

To alleviate both of these problems, we consider a *linear connector*, *i.e.*, set $\frac{\partial \hat{z}_i}{\partial \underline{z}_i} = \frac{\overline{z}_i - \hat{z}_i}{\overline{z}_i - \underline{z}_i}$. However, even when our latent adversarial example is very close to one bound, the linear connector would induce non-zero gradients w.r.t. to the opposite bound. To remedy this undesirable behavior, we propose the *rectified linear connector*, setting $\frac{\partial \hat{z}_i}{\partial \underline{z}_i} = \max(0, 1 - \frac{\hat{z}_i - \underline{z}_i}{c(\overline{z}_i - \underline{z}_i)})$ where $c \in [0, 1]$ is a constant (visualized in Figure 3 for $c = 0.3$). Observe that it recovers the binary connector

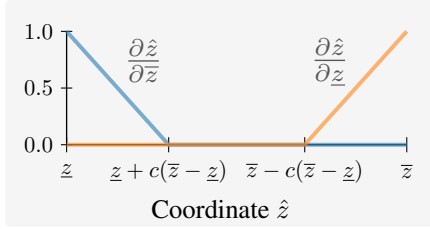

Figure 3: Gradient connector visualization.

for $c = 0$ and the linear connector for $c = 1$. To prevent gradient sparsity ($c \leq 0.5$) while avoiding the above-mentioned counterintuitive gradient connections ($c \geq 0.5$), we set $c = 0.5$ unless indicated otherwise. When the upper and lower bounds are identical in the $i^{\text{th}}$ dimension, PGD turns into an identity function. Therefore, we set both gradients to $\frac{\partial \hat{z}_i}{\partial \underline{z}_i} = \frac{\partial \hat{z}_i}{\partial \overline{z}_i} = 0.5$ turning the gradient connector into an identity function for the backward pass.

### 3.3 TAPS Loss & Multi-estimator PGD

The standard PGD attack, used in adversarial training, henceforth called *single-estimator* PGD, is based on maximizing the Cross-Entropy loss $\mathcal{L}_{\text{CE}}$ of a single input. In the context of TAPS, this results in the overall loss

$$\mathcal{L}_{\text{TAPS}}^{\text{single}}(\boldsymbol{x}, y, \epsilon) = \max_{\hat{\boldsymbol{z}} \in [\underline{z}, \overline{z}]} \ln\left(1 + \sum_{i \neq y} \exp(f_C(\hat{\boldsymbol{z}})_i - f_C(\hat{\boldsymbol{z}})_y)\right),$$

where the embedding space bounding box $[\underline{z}, \overline{z}]$ is obtained via IBP. However, this loss is not necessarily well aligned with adversarial robustness. Consider the example illustrated in Figure 4, where only points in the lower-left quadrant are classified correctly (*i.e.*, $o_i^\Delta := o_i - o_y < 0$). We compute the latent adversarial example $\hat{z}$ by conducting a standard adversarial attack on the Cross-Entropy loss over the reachable set ■ (optimally for illustration purposes) and observe that the

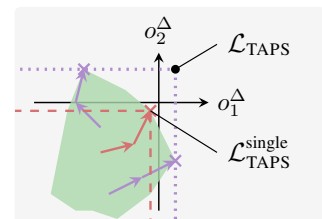

Figure 4: Illustration of the bounds on $o_i^\Delta := o_i - o_t$ obtained via single estimator (- -) and multi-estimator (⋯) PGD and the points maximizing the corresponding losses: × for $\mathcal{L}_{\text{TAPS}}^{\text{single}}$ and • for $\mathcal{L}_{\text{TAPS}}$.

corresponding output $\boldsymbol{f}(\hat{\boldsymbol{z}})$ (×) is classified correctly. However, if we instead use the logit differences $o_1^\Delta$ and $o_2^\Delta$ as attack objectives, we obtain two misclassified points (×). Combining their dimension-wise worst-case bounds (⋯), we obtain the point •, which realizes the maximum loss over an optimal box approximation of the reachable set. As the correct classification of this point (when computed exactly) directly corresponds to true robustness, we propose the *multi-estimator* PGD variant of $\mathcal{L}_{\text{TAPS}}$, which estimates the upper bounds on the logit differences $o_i^\Delta$ using separate samples and then computes the loss function using the per-dimension worst-cases as:

$$\mathcal{L}_{\text{TAPS}}(\boldsymbol{x}, y, \epsilon) = \ln\left(1 + \sum_{i \neq y} \exp\left(\max_{\hat{\boldsymbol{z}} \in [\underline{z}, \overline{z}]} f_C(\hat{\boldsymbol{z}})_i - f_C(\hat{\boldsymbol{z}})_y\right)\right).$$

### 3.4 Training Objective & Regularization

While complete certification methods can decide any robustness property, this requires exponential time. Therefore, networks should not only be robust but also certifiable. Thus, we propose to combine the IBP loss for easy-to-learn and certify samples with the TAPS loss for harder samples as follows:

$$\mathcal{L}(\boldsymbol{x}, y, \epsilon) = \mathcal{L}_{\text{TAPS}}(\boldsymbol{x}, y, \epsilon) \cdot \mathcal{L}_{\text{IBP}}(\boldsymbol{x}, y, \epsilon).$$

This expresses that every sample should be either certifiable with TAPS or IBP bounds[1]. Further, as by construction $\mathcal{L}_{\text{TAPS}} \le \mathcal{L}_{\text{IBP}}$, we add a scaling term $\alpha$ to the loss gradient:

$$\frac{d\mathcal{L}}{d\boldsymbol{\theta}} := 2\alpha \frac{d\mathcal{L}_{\text{TAPS}}}{d\boldsymbol{\theta}} \cdot \mathcal{L}_{\text{IBP}} + (2 - 2\alpha) \frac{d\mathcal{L}_{\text{IBP}}}{d\boldsymbol{\theta}} \cdot \mathcal{L}_{\text{TAPS}}.$$

Here, $\alpha = 0.5$ recovers the standard gradient, obtained via the product rule (both sides weighted with 1), while $\alpha = 0$ and $\alpha = 1$ correspond to using only the (weighted) IBP and TAPS gradients, respectively. Henceforth, we express this as the regularization weight $w_{\text{TAPS}} = \frac{\alpha}{1-\alpha}$, which intuitively expresses the weight put on TAPS, using $w_{\text{TAPS}} = 5$ unless specified otherwise. Lastly, we reduce the variance of $\mathcal{L}$ by averaging $\mathcal{L}_{\text{IBP}}$ and $\mathcal{L}_{\text{TAPS}}$ over a mini batch before multiplying (see App. A).

### 3.5 STAPS – Balancing Regularization by Combining TAPS with SABR

Recall that SABR (Müller et al., 2022a) reduces the over-regularization of certified training by propagating a small, adversarially selected BOX through the whole network. However, as BOX approximations grow exponentially with depth (Müller et al., 2022a; Shi et al., 2021; Mao et al., 2023), regardless of the input region size, SABR has to strike a balance between regularizing early layers too little and later layers too much. In contrast, TAPS's approach of propagating the full input region through the first part of the network (the feature extractor) before using PGD for the remainder reduces regularization only in later layers. Thus, we propose STAPS by replacing the IBP components of TAPS, for both propagation and regularization, with SABR to obtain a more uniform reduction of over-regularization throughout the whole network.

STAPS, identically to SABR, first conducts a PGD attack over the whole network to find an adversarial example $\boldsymbol{x}' \in \mathcal{B}(\boldsymbol{x}', \epsilon - \tau)$. Then, it propagates BOX-bounds for a small region $\mathcal{B}(\boldsymbol{x}', \tau) \subset \mathcal{B}(\boldsymbol{x}, \epsilon)$ (with $\tau < \epsilon$) around this adversarial example $\boldsymbol{x}'$ through the feature extractor, before, identically to TAPS, conducting an adversarial attack in the resulting latent-space region over the classifier component of the network.

## 4 Experimental Evaluation

In this section, we evaluate TAPS empirically, first, comparing it to a range of state-of-the-art certified training methods, before conducting an extensive ablation study validating our design choices.

**Experimental Setup** We implement TAPS in PyTorch (Paszke et al., 2019) and use MN-BAB (Ferrari et al., 2022) for certification. We conduct experiments on MNIST (LeCun et al., 2010), CIFAR-10 (Krizhevsky et al., 2009), and TINYIMAGENET (Le and Yang, 2015) using $\ell_\infty$ perturbations and the CNN7 architecture (Gowal et al., 2018). For more experimental details including hyperparameters and computational costs and an extended analysis see App. B and App. C, respectively.

### 4.1 Main Results

In Table 1, we compare TAPS to state-of-the-art certified training methods. Most closely related are IBP, recovered by TAPS if the classifier size is zero, and COLT, which also combines bound propagation with adversarial attacks but does not allow for joint training. TAPS dominates IBP, improving on its certified and natural accuracy in all settings and demonstrating the importance of avoiding over-regularization. Compared to COLT, TAPS improves certified accuracies significantly, highlighting the importance of joint optimization. In some settings, this comes at the cost of slightly reduced natural accuracy, potentially due to COLT's use of the more precise ZONOTOPE approximations. Compared to the recent SABR and IBP-R, TAPS often achieves higher certified accuracies at

---

[1]See Fischer et al. (2019) for further discussion.

Table 1: Comparison of natural (Nat.) and certified (Cert.) accuracy on the full MNIST, CIFAR-10, and TINYIMAGENET test sets. We report results for other methods from the relevant literature.

| Dataset | $\epsilon_\infty$ | Training Method | Source | Nat. [%] | Cert. [%] |
|---|---|---|---|---|---|
| MNIST | 0.1 | COLT | Balunovic and Vechev (2020) | 99.2 | 97.1 |
| | | IBP | Shi et al. (2021) | 98.84 | 97.95 |
| | | SORTNET | Zhang et al. (2022b) | 99.01 | 98.14 |
| | | SABR | Müller et al. (2022a) | **99.23** | 98.22 |
| | | TAPS | this work | 99.19 | **98.39** |
| | | STAPS | this work | 99.15 | 98.37 |
| | 0.3 | COLT | Balunovic and Vechev (2020) | 97.3 | 85.7 |
| | | IBP | Shi et al. (2021) | 97.67 | 93.10 |
| | | SORTNET | Zhang et al. (2022b) | 98.46 | 93.40 |
| | | SABR | Müller et al. (2022a) | **98.75** | 93.40 |
| | | TAPS | this work | 97.94 | **93.62** |
| | | STAPS | this work | 98.53 | 93.51 |
| CIFAR-10 | $\frac{2}{255}$ | COLT | Balunovic and Vechev (2020) | 78.4 | 60.5 |
| | | IBP | Shi et al. (2021) | 66.84 | 52.85 |
| | | SORTNET | Zhang et al. (2022b) | 67.72 | 56.94 |
| | | IBP-R | Palma et al. (2022) | 78.19 | 61.97 |
| | | SABR | Müller et al. (2022a) | 79.24 | 62.84 |
| | | TAPS | this work | 75.09 | 61.56 |
| | | STAPS | this work | **79.76** | **62.98** |
| | $\frac{8}{255}$ | COLT | Balunovic and Vechev (2020) | 51.7 | 27.5 |
| | | IBP | Shi et al. (2021) | 48.94 | 34.97 |
| | | SORTNET | Zhang et al. (2022b) | **54.84** | **40.39** |
| | | IBP-R | Palma et al. (2022) | 51.43 | 27.87 |
| | | SABR | Müller et al. (2022a) | 52.38 | 35.13 |
| | | TAPS | this work | 49.76 | 35.10 |
| | | STAPS | this work | 52.82 | 34.65 |
| TINYIMAGENET | $\frac{1}{255}$ | IBP | Shi et al. (2021) | 25.92 | 17.87 |
| | | SORTNET | Zhang et al. (2022b) | 25.69 | 18.18 |
| | | SABR | Müller et al. (2022a) | 28.85 | 20.46 |
| | | TAPS | this work | 28.34 | 20.82 |
| | | STAPS | this work | **28.98** | **22.16** |

the cost of slightly reduced natural accuracies. Reducing regularization more uniformly with STAPS achieves higher certified accuracies in almost all settings and better natural accuracies in many, further highlighting the orthogonality of TAPS and SABR. Most notably, STAPS increases certified accuracy on TINYIMAGENET by almost 10% while also improving natural accuracy. SORTNET, a generalization of a range of recent architectures (Zhang et al., 2021, 2022c; Anil et al., 2019), introducing novel activation functions tailored to yield networks with high $\ell_\infty$-robustness, performs well on CIFAR-10 at $\epsilon = 8/255$, but is dominated by STAPS in every other setting.

### 4.2 Ablation Study

**Approximation Precision** To evaluate whether TAPS yields more precise approximations of the worst-case loss than other certified training methods, we compute approximations of the maximum margin loss with IBP, PGD (50 steps, 3 restarts), SABR ($\lambda = 0.4$), and TAPS on a small TAPS-trained CNN3 for all MNIST test set samples. We report histograms over the difference to the exact worst-case loss computed with a MILP encoding (Tjeng et al., 2019) in Figure 5. Positive values correspond to over-approximations while negative values correspond to under-approximation. We observe that the TAPS approximation is by far the most precise, achieving the smallest mean and mean absolute error as well as variance. We confirm these observations for other training methods in Figure 9 in App. C.

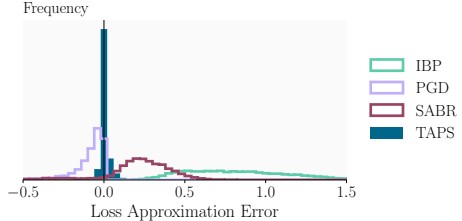

Figure 5: Distribution of the worst-case loss approximation errors over test set samples.

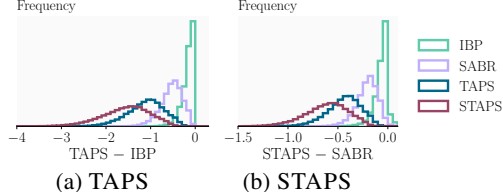

(a) TAPS          (b) STAPS

Figure 6: Bound difference between IBP and PGD propagation through the classifier depending on the training method.

To isolate the under-approximation effect of the PGD propagation through the classifier, we visualize the distribution over pairwise bound differ-

ences between TAPS and IBP and STAPS and SABR in Figure 6 for different training methods. We observe that the distributions for TAPS and STAPS are remarkably similar (up to scaling), highlighting the importance of reducing over-regularisation of the later layers, even when propagating only small regions (SABR/STAPS). Further, we note that larger bound differences indicate reduced regularisation of the later network layers. We thus observe that SABR still induces a much stronger regularisation of the later layers than TAPS and especially STAPS, again highlighting the complementarity of TAPS and SABR, discussed in Section 3.5.

**IBP Regularization** To analyze the effectiveness of the multiplicative IBP regularization discussed in Section 3.4, we train with IBP in isolation ($\mathcal{L}_{\text{IBP}}$), IBP with TAPS weighted gradients ($w_{\text{TAPS}} = 0$), varying levels of gradient scaling for the TAPS component ($w_{\text{TAPS}} \in [1, 20]$), TAPS with IBP weighting ($w_{\text{TAPS}} = \infty$), and TAPS loss in isolation, reporting results in Table 2. We observe that IBP in isolation yields comparatively low standard but moderate certified accuracies with fast certification times. Increasing the weight $w_{\text{TAPS}}$ of the TAPS gradients reduces regu-

Table 2: Effect of IBP regularization and the TAPS gradient expanding coefficient $\alpha$ for MNIST $\epsilon = 0.3$.

| $w_{\text{TAPS}}$ | Avg time (s) | Nat (%) | Adv. (%) | Cert. (%) |
|---|---|---|---|---|
| $\mathcal{L}_{\text{IBP}}$ | 2.3 | 97.6 | 93.37 | 93.15 |
| 0 | 2.7 | 97.37 | 93.32 | 93.06 |
| 1 | 4.5 | 97.86 | 93.80 | 93.36 |
| 5 | 6.9 | 97.94 | 94.01 | **93.62** |
| 10 | 15.7 | 98.25 | 94.43 | 93.02 |
| 15[†] | 42.8 | 98.53 | **95.00** | 91.55 |
| 20[†] | 73.7 | **98.75** | 94.33 | 82.67 |
| $\infty$[†] | 569.7 | 98.0 | 94.00 | 45.00 |
| $\mathcal{L}_{\text{TAPS}}$[†] | 817.1 | 98.5 | 94.50 | 17.50 |

[†] Only evaluated on part of the test set within a 2-day time limit.

larization, leading to longer certification times and higher standard accuracies. Initially, this translates to higher adversarial and certified accuracies, peaking at $w_{\text{TAPS}} = 15$ and $w_{\text{TAPS}} = 5$, respectively, before especially certified accuracy decreases as regularization becomes insufficient for certification. We confirm these trends for TINYIMAGENET in Table 14 in App. C.

**Split Location** TAPS splits a given network into a feature extractor and classifier, which are then approximated using IBP and PGD, respectively. As IBP propagation accumulates over-approximation errors while PGD is an under-approximation, the location of this split has a strong impact on the regularization level induced by TAPS. To analyze this effect, we train multiple

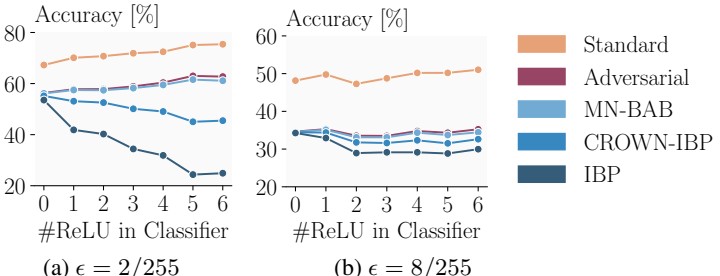

(a) $\epsilon = 2/255$      (b) $\epsilon = 8/255$

Figure 7: Effect of split location on the standard and robust accuracy of TAPS trained networks, depending on the perturbation magnitude $\epsilon$ for different certification methods for CIFAR-10. 0 ReLUs in the classifier recovers IBP training.

CNN7s such that we obtain classifier components with between 0 and 6 (all) ReLU layers and illustrate the resulting standard, adversarial, and certified (using different methods) accuracies in Figure 7 for CIFAR-10, and in App. C for MNIST and TINYIMAGENET in Tables 11 and 13 respectively.

For small perturbations ($\epsilon = 2/255$), increasing classifier size and thus decreasing regularization yields increasing natural and adversarial accuracy. While the precise MN-BAB verification can translate this to rising certified accuracies up to large classifier sizes, regularization quickly becomes insufficient for the less precise IBP and CROWN-IBP certification. For larger perturbations ($\epsilon = 8/255$), the behavior is more complex. An initial increase of all accuracies with classifier size is followed by a sudden drop and slow recovery, with certified accuracies remaining below the level achieved for 1 ReLU layer. We hypothesize that this effect is due to the IBP regularization starting to dominate optimization combined with increased training difficulty (see App. C for details). For both perturbation magnitudes, gains in certified accuracy can only be realized with the precise MN-BAB certification (Müller et al., 2022a), highlighting the importance of recent developments in neural network verification for certified training.

**Gradient Connector**   In Figure 8, we illustrate the effect of our gradient connector's parameterization c (Section 3.2). We report TAPS accuracy (the portion of samples where all latent adversarial examples are classified correctly) as a proxy for the goodness of fit. Recall that $c = 0$ corresponds to the binary connector and $c = 1$ to the linear connector. We observe that the binary connector achieves poor TAPS and natural accuracy, indicating a less well-behaved optimization problem. TAPS accuracy peaks at $c = 0.5$, indicating

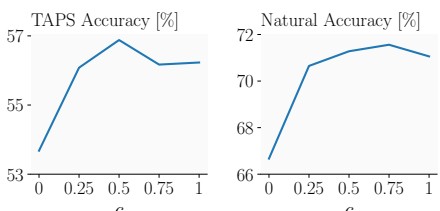

Figure 8: Effect of the gradient connector on TAPS (left) and natural (right) accuracy.

high goodness-of-fit and thus a well-behaved optimization problem. This agrees well with our theoretical considerations aiming to avoid sparsity ($c < 0.5$) and contradicting gradients ($c > 0.5$).

**Single-Estimator vs Multi-Estimator PGD** To evaluate the importance of our multi-estimator PGD variant, we compare it to single-estimator PGD across a range of split positions, reporting results in Table 3. We observe that across all split positions, multi-estimator PGD achieves better certified and better or equal natural accuracy. Further, training collapses reproducibly for single-estimator PGD for small classifiers, indicating that multi-estimator PGD additionally improves training stability.

Table 3: Comparison of single- and multi-estimator PGD, depending on the split position for MNIST at $\epsilon = 0.3$.

| # ReLU in Classifier | Single | | Multi | |
|---|---|---|---|---|
| | Certified | Natural | Certified | Natural |
| 1 | -[†] | 31.47[†] | **93.62** | 97.94 |
| 3 | 92.91 | 98.56 | 93.03 | 98.63 |
| 6 | 92.41 | **98.88** | 92.70 | **98.88** |

[†] Training encounters mode collapse. Last epoch performance reported.

**PGD Attack Strength** To investigate the effect of the adversarial attack's strength, we use 1 or 3 restarts and vary the number of attack steps used in TAPS from 1 to 100 for MNIST at $\epsilon = 0.3$, reporting results in Table 4. Interestingly, even a single attack step and restart are sufficient to achieve good performance and outperform IBP. As we increase the strength of the attack, we can increase certified accuracy slightly while marginally reducing natural accuracy, agreeing well with our expectation of regularization strength increasing with attack strength.

Table 4: Effect of different PGD attack strengths for MNIST at $\epsilon = 0.3$.

| # Attack Steps | 1 Restart | | 3 Restarts | |
|---|---|---|---|---|
| | Certified | Natural | Certified | Natural |
| 1 | 93.36 | **98.22** | 93.47 | **98.22** |
| 5 | 93.15 | 97.90 | **93.55** | 97.90 |
| 20 | **93.62** | 97.94 | 93.52 | 97.99 |
| 100 | 93.46 | 97.94 | **93.55** | 97.99 |

## 5   Related Work

**Verification Methods**   In this work, we only consider deterministic verification methods, which analyze a given network as is. While *complete* (or *exact*) methods (Tjeng et al., 2019; Wang et al., 2021; Zhang et al., 2022a; Ferrari et al., 2022) can decide any robustness property given enough time, *incomplete* methods (Singh et al., 2018; Raghunathan et al., 2018; Zhang et al., 2018; Dathathri et al., 2020; Müller et al., 2022b) sacrifice some precision for better scalability. However, recent complete methods can be used with a timeout to obtain effective incomplete methods.

**Certified Training**   Most certified training methods compute and minimize sound over-approximations of the worst-case loss using different approximation methods: DIFFAI (Mirman et al., 2018) and IBP (Gowal et al., 2018) use BOX approximations, Wong et al. (2018) use DEEPZ relaxations (Singh et al., 2018), Wong and Kolter (2018) back-substitute linear bounds using fixed relaxations, Zhang et al. (2020) use dynamic relaxations (Zhang et al., 2018; Singh et al., 2019a) and compute intermediate bounds using BOX relaxations. Shi et al. (2021) significantly shorten training schedules by combining IBP training with a special initialization. Some more recent methods instead compute and optimize more precise, but not necessarily sound, worst-case loss approximations: SABR (Müller et al., 2022a) reduce the regularization of IBP training by propagating only small but carefully selected subregions. IBP-R (Palma et al., 2022) combines adversarial training at large perturbation radii with an IBP-based regularization. COLT (Balunovic and Vechev, 2020) is conceptually most similar to TAPS and thus compared to in more detail below. While prior work

combined a robust and a precise network (Müller et al., 2021; Horváth et al., 2022a), to trade-off certified and standard accuracy, these unsound certified training methods can often increase both.

COLT (Balunovic and Vechev, 2020), similar to TAPS, splits the network into a feature extractor and classifier, computing bounds on the feature extractor's output (using the ZONOTOPE (Singh et al., 2019a) instead of BOX domain) before conducting adversarial training over the resulting region. Crucially, however, COLT lacks a gradient connector and, thus, does not enable gradient flow between the latent adversarial examples and the bounds on the feature extractor's output. Therefore, gradients can only be computed for the weights of the classifier but not the feature extractor, preventing the two components from being trained jointly. Instead, a stagewise training process is used, where the split between feature extractor and classifier gradually moves through the network starting with the whole network being treated as the classifier. This has several repercussions: not only is the training very slow and limited to relatively small networks (a four-layer network takes almost 2 days to train) but more importantly, the feature extractor (and thus the whole network) is never trained specifically for precise bound propagation. Instead, only the classifier is trained to become robust to the incurred imprecisions. As this makes bound propagation methods ineffective for certification, Balunovic and Vechev (2020) employ precise but very expensive mixed integer linear programming (MILP (Tjeng et al., 2019)), further limiting the scalability of COLT.

In our experimental evaluation (Section 4.1), we compare TAPS in detail to the above methods.

**Robustness by Construction** Li et al. (2019), Lécuyer et al. (2019), and Cohen et al. (2019) construct probabilistic classifiers by introducing randomness into the inference process of a base classifier. This allows them to derive robustness guarantees with high probability at the cost of significant (100x) runtime penalties. Salman et al. (2019) train the base classifier using adversarial training and Horváth et al. (2022b) ensemble multiple base models to improve accuracies at a further runtime penalty. Zhang et al. (2021, 2022c) introduce $\ell_\infty$-distance neurons, generalized to SORTNET by Zhang et al. (2022b) which inherently exhibits $\ell_\infty$-Lipschitzness properties, yielding good robustness for large perturbation radii, but poor performance for smaller ones.

# 6 Conclusion

We propose TAPS, a novel certified training method that reduces over-regularization by constructing and optimizing a precise worst-case loss approximation based on a combination of IBP and PGD training. Crucially, TAPS enables joint training over the IBP and PGD approximated components by introducing the gradient connector to define a gradient flow through their interface. Empirically, we confirm that TAPS yields much more precise approximations of the worst-case loss than existing methods and demonstrate that this translates to state-of-the-art performance in certified training in many settings.

# Acknowledgements

We would like to thank our anonymous reviewers for their constructive comments and insightful questions.

This work has been done as part of the EU grant ELSA (European Lighthouse on Secure and Safe AI, grant agreement no. 101070617) and the SERI grant SAFEAI (Certified Safe, Fair and Robust Artificial Intelligence, contract no. MB22.00088). Views and opinions expressed are however those of the authors only and do not necessarily reflect those of the European Union or European Commission. Neither the European Union nor the European Commission can be held responsible for them.

The work has received funding from the Swiss State Secretariat for Education, Research and Innovation (SERI).

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

## A  Averaging Multipliers Makes Gradients Efficient

**Theorem 1.** *Let $x_i$ be i.i.d. drawn from the dataset and define $f_i = f_\theta(x_i)$ and $g_i = g_\theta(x_i)$, where $f_\theta$ and $g_\theta$ are two functions. Further, define $L_1 = (\sum_{i=1}^n \frac{1}{n} f_i) \cdot (\sum_{i=1}^n \frac{1}{n} g_i)$ and $L_2 = \sum_{i=1}^n \frac{1}{n} f_i g_i$. Then, assuming the function value and the gradient are independent, $\mathbb{E}_x \left( \frac{\partial L_1}{\partial \theta} \right) = \mathbb{E}_x \left( \frac{\partial L_2}{\partial \theta} \right)$ and $\mathrm{Var}_x \left( \frac{\partial L_1}{\partial \theta} \right) \leq \mathrm{Var}_x \left( \frac{\partial L_2}{\partial \theta} \right)$.*

*Proof.* A famous result in stochastic optimization is that stochastic gradients are unbiased. For completeness, we give a short proof of this property: Let $L = \mathbb{E}_x f(x) = \int_{-\infty}^{+\infty} f(x) dP(x)$, thus $\nabla_x L = \nabla_x (\int_{-\infty}^{+\infty} f(x) dP(x)) = \int_{-\infty}^{+\infty} \nabla_x f(x) dP(x) = \mathbb{E}_x(\nabla_x f(x))$. Therefore, $\nabla f(x_i)$ is an unbiased estimator of the true gradient.

Applying that the stochastic gradients are unbiased, we can write $\nabla_\theta f_i = \nabla_\theta f + \eta_i$, where $\nabla_\theta f$ is the expectation of the gradient and $\eta_i$ is the deviation such that $\mathbb{E}\eta_i = 0$ and $\mathrm{Var}(\eta_i) = \sigma_1^2$. Since $x_i$ is drawn independently, $f_i$ are independent and thus $\eta_i$ are independent. Similarly, we can write $\nabla_\theta g_i = \nabla_\theta g + \delta_i$, where $\mathbb{E}\delta_i = 0$ and $\mathrm{Var}(\delta_i) = \sigma_2^2$. $\eta_i$ and $\delta_i$ may be dependent.

Define $\bar{f} = \sum_i \frac{1}{n} f_i$ and $\bar{g} = \sum_i \frac{1}{n} g_i$. Explicit computation gives us that $\nabla L_1 = \bar{g} \cdot \left( \sum_i \frac{1}{n} \nabla f_i \right) + \bar{f} \cdot \left( \sum_i \frac{1}{n} \nabla g_i \right)$, and $\nabla L_2 = \sum_i \frac{1}{n} (f_i \nabla g_i + g_i \nabla f_i)$. Therefore,

$$\mathbb{E}_x \left( \nabla_\theta L_1 \mid f_i, g_i \right) = \bar{g} \nabla_\theta f + \bar{f} \nabla_\theta g = \mathbb{E}_x \left( \nabla_\theta L_2 \mid f_i, g_i \right).$$

By the law of total probability,

$$\begin{aligned}
\mathbb{E}_x \left( \nabla_\theta L_1 \right) &= \mathbb{E}_{f_i, g_i} \left( \mathbb{E}_x \left( \nabla_\theta L_1 \mid f_i, g_i \right) \right) \\
&= \mathbb{E}_{f_i, g_i} \left( \mathbb{E}_x \left( \nabla_\theta L_2 \mid f_i, g_i \right) \right) \\
&= \mathbb{E}_x \left( \nabla_\theta L_2 \right).
\end{aligned}$$

Therefore, we have got the first result: the gradients of $L_1$ and $L_2$ have the same expectation.

To prove the variance inequality, we will use variance decomposition formula[2]:

$$\begin{aligned}
\mathrm{Var}_x(\nabla_\theta L_k) = {} & \mathbb{E}_{f_i, g_i}(\mathrm{Var}_x(\nabla_\theta L_k \mid f_i, g_i)) + \\
& \mathrm{Var}_{f_i, g_i}(\mathbb{E}_x(\nabla_\theta L_k \mid f_i, g_i)),
\end{aligned}$$

$k = 1, 2$. We have proved that $\mathbb{E}_x(\nabla_\theta L_1 \mid f_i, g_i) = \mathbb{E}_x(\nabla_\theta L_2 \mid f_i, g_i)$, thus the second term is equal. Next, we prove that $\mathrm{Var}_x(\nabla_\theta L_1 \mid f_i, g_i) \leq \mathrm{Var}_x(\nabla_\theta L_2 \mid f_i, g_i)$, which implies $\mathrm{Var}_x(\nabla_\theta L_1) \leq \mathrm{Var}_x(\nabla_\theta L_2)$.

By explicit computation, we have

$$\begin{aligned}
& \mathrm{Var}(\nabla L_1 \mid f_i, g_i) \\
& = (\bar{g})^2 \mathrm{Var} \left( \sum_i \frac{1}{n} \eta_i \right) + (\bar{f})^2 \mathrm{Var} \left( \sum_i \frac{1}{n} \delta_i \right) \\
& = \frac{1}{n} \sigma_1^2 (\bar{g})^2 + \frac{1}{n} \sigma_2^2 (\bar{f})^2,
\end{aligned} \tag{4}$$

and

$$\begin{aligned}
& \mathrm{Var}(\nabla L_2 \mid f_i, g_i) \\
& = \mathrm{Var} \left( \sum_i \frac{1}{n} f_i \delta_i \right) + \mathrm{Var} \left( \sum_i \frac{1}{n} g_i \eta_i \right) \\
& = \frac{1}{n} \sigma_1^2 \left( \sum_i \frac{1}{n} g_i^2 \right) + \frac{1}{n} \sigma_2^2 \left( \sum_i \frac{1}{n} f_i^2 \right).
\end{aligned} \tag{5}$$

Applying Jensen's formula on the convex function $x^2$, we have $\left( \sum_i \frac{1}{n} a_i \right)^2 \leq \sum_i \frac{1}{n} a_i^2$ for any $a_i$, thus $(\bar{f})^2 \leq \sum_i \frac{1}{n} f_i^2$ and $(\bar{g})^2 \leq \sum_i \frac{1}{n} g_i^2$. Combining Equation (4) and Equation (5) with these two inequalities gives the desired result. $\qquad \square$

---

[2] https://en.wikipedia.org/wiki/Law_of_total_variance

**Algorithm 1** Train Loss Computation

---

**Input:** data $X_B = \{(\boldsymbol{x}_b, y_b)\}_b$, current $\epsilon$, target $\epsilon^t$, network $\boldsymbol{f}$
**Output:** A differentiable loss $L$
$\mathcal{L}_{\text{IBP}} = \sum_{b \in \mathcal{B}} \mathcal{L}_{\text{IBP}}(\boldsymbol{x}_b, y_b, \epsilon)/|\mathcal{B}|$.
**if** $\epsilon < \epsilon^t$ **then**
    // $\epsilon$ annealing regularisation from Shi et al. (2021)
    $\mathcal{L}_{\text{fast}} = \lambda \cdot (\mathcal{L}_{\text{tightness}} + \mathcal{L}_{\text{relu}})$
    **return** $\mathcal{L}_{\text{IBP}} + \epsilon/\epsilon^t \cdot \mathcal{L}_{\text{fast}}$
$\mathcal{L}_{\text{TAPS}} = \sum_{b \in \mathcal{B}} L_{\text{TAPS}}(\boldsymbol{x}_b, y_b, \epsilon)/|\mathcal{B}|$.
**return** $\mathcal{L}_{\text{IBP}} \cdot \mathcal{L}_{\text{TAPS}}$

---

Table 5: The training epoch and learning rate settings.

| Dataset | Batch size | Total epochs | Annealing epochs | Decay-1 | Decay-2 |
|---|---|---|---|---|---|
| MNIST | 256 | 70 | 20 | 50 | 60 |
| CIFAR-10 | 128 | 160 | 80 | 120 | 140 |
| TINYIMAGENET | 128 | 80 | 20 | 60 | 70 |

## B Experiment Details

### B.1 TAPS Training Procedure

To obtain state-of-the-art performance with IBP, various training techniques have been developed. We use two of them: $\epsilon$-annealing (Gowal et al., 2018) and initialization and regularization for stable box sizes (Shi et al., 2021). $\epsilon$-annealing slowly increases the perturbation magnitude $\epsilon$ during training to avoid exploding approximation sizes and thus gradients. The initialization of Shi et al. (2021) scales network weights to achieve constant box sizes over network depth. During the $\epsilon$-annealing phase, we combine the IBP loss with the ReLU stability regularization $\mathcal{L}_{\text{fast}}$ (Shi et al., 2021), before switching to the TAPS loss as described in Section 3.4. We formalize this in Algorithm 1. We follow Shi et al. (2021) in doing early stopping based on validation set performance. However, we use TAPS accuracy (see App. C) instead of IBP accuracy as a performance metric.

### B.2 Datasets and Augmentation

We use the MNIST (LeCun et al., 2010), CIFAR-10 (Krizhevsky et al., 2009), and TINYIMAGENET (Le and Yang, 2015) datasets, all of which are freely available with no license specified.

The data preprocessing mostly follows Müller et al. (2022a). For MNIST, we do not apply any preprocessing. For CIFAR-10 and TINYIMAGENET, we normalize with the dataset mean and standard deviation (after calculating perturbation size) and augment with random horizontal flips. For CIFAR-10, we apply random cropping to $32 \times 32$ after applying a 2 pixel padding at every margin. For TINYIMAGENET, we apply random cropping to $56 \times 56$ during training and center cropping during testing.

### B.3 Model Architectures

Unless specified otherwise, we follow Shi et al. (2021); Müller et al. (2022a) and use a `CNN7` with Batch Norm for our main experiments. `CNN7` is a convolutional network with 7 convolutional and linear layers. All but the last linear layer are followed by a Batch Norm and ReLU layer.

### B.4 Training Hyperparameter Details

We follow the hyperparameter choices of Shi et al. (2021) for $\epsilon$-annealing, learning rate schedules, batch sizes, and gradient clipping (see Table 5). We set the initial learning rate to 0.0005 and decrease it by a factor of 0.2 at Decay-1 and -2. We set the gradient clipping threshold to 10.

We use additional $L_1$ regularization in some settings where we observe signs of overfitting. We report the $L_1$ regularization and split position chosen for different settings in Table 6 and Table 8.

Table 6: Hyperparameters for TAPS.

| Dataset | $\epsilon$ | # ReLUs in Classifier | $L_1$ | $w$ |
|---|---|---|---|---|
| MNIST | 0.1 | 3 | 1e-6 | 5 |
| | 0.3 | 1 | 0 | 5 |
| CIFAR-10 | 2/255 | 5 | 2e-6 | 5 |
| | 8/255 | 1 | 2e-6 | 5 |
| TINYIMAGENET | 1/255 | 1 | 0 | 5 |

Table 7: Training and certification times for TAPS-trained networks.

| Dataset | $\epsilon$ | Train Time (s) | Certify Time (s) |
|---|---|---|---|
| MNIST | 0.1 | 42 622 | 17 117 |
| | 0.3 | 12 417 | 41 624 |
| CIFAR-10 | 2/255 | 141 281 | 166 474 |
| | 8/255 | 27 017 | 26 968 |
| TINYIMAGENET | 1/255 | 306 036 | 23 497 |

We train using single NVIDIA GeForce RTX 3090 for MNIST and CIFAR-10 and single NVIDIA TITAN RTX for TINYIMAGENET. Training and certification times are reported in Table 7 and Table 9.

### B.5 Certification Details

We combine IBP (Gowal et al., 2018), CROWN-IBP (Zhang et al., 2020), and MN-BAB (Ferrari et al., 2022) for certification, running the most precise but also computationally costly MN-BAB only on samples not certified by the other methods. We use the same configuration for MN-BAB as Müller et al. (2022a). The certification is run on a single NVIDIA TITAN RTX.

MN-BAB Ferrari et al. (2022) is a state-of-the-art (Brix et al., 2023; Müller et al., 2022c) neural network verifier, combining the branch-and-bound paradigm (Bunel et al., 2020) with precise multi-neuron constraints (Müller et al., 2022b; Singh et al., 2019b).

We use a mixture of strong adversarial attacks to evaluate adversarial accuracy. First, we run PGD attacks with 5 restarts and 200 iterations each. Then, we run MN-BaB to search for adversarial examples with a timeout of 1000 seconds.

## C   Extended Evaluation

**TAPS Accuracy as GoF**   In practice, we want to avoid certifying every model with expensive certification methods, especially during hyperparameter tuning and for early stopping. Therefore, we need a criterion to select models. In this section, we aim to show that TAPS accuracy (accuracy of the latent adversarial examples) is a good proxy for goodness of fit (GoF).

We compare the TAPS accuracy to adversarial and certified accuracy with all models we get on MNIST and CIFAR-10. The result is shown in Table 10. We can see that the correlations between TAPS accuracy and both the adversarial and the certified accuracy are close to 1. In addition, the differences are small and centered at zero, with a small standard deviation. Therefore, we conclude that TAPS accuracy is a good estimate of the true robustness, thus a good measurement of GoF. In all the experiments, we perform model selection based on the TAPS accuracy.

**Training Difficulty**   Since TAPS is merely a training technique, we can test TAPS-trained models using a different classifier split. By design, if the training is successful, then under a given classifier split for testing, the model trained with the same split should have the best TAPS accuracy. Although this is often true, we find that in some cases, a smaller classifier split results in higher TAPS accuracy, indicating optimization issues.

We measure TAPS accuracy for models trained with IBP and TAPS using different splits for CIFAR-10 (Figure 10) and MNIST (Figure 11). We observe that for CIFAR-10 $\epsilon = 2/255$ and MNIST, the models trained and tested with the classifier/extractor split achieve the highest TAPS accuracies, as expected, indicating a relatively well-behaved optimization problem. However, for CIFAR-10 $\epsilon = 8/255$, the model trained with a classifier size of 1 achieves the highest TAPS accuracy for all

Table 8: Hyperparameter for STAPS.

| Dataset | $\epsilon$ | # ReLUs in Classifier | $L_1$ | $w$ | $\tau/\epsilon$ |
|---|---|---|---|---|---|
| MNIST | 0.1 | 1 | 2e-5 | 5 | 0.4 |
| | 0.3 | 1 | 2e-6 | 5 | 0.6 |
| CIFAR-10 | 2/255 | 1 | 2e-6 | 2 | 0.1 |
| | 8/255 | 1 | 2e-6 | 5 | 0.7 |
| TINYIMAGENET | 1/255 | 2 | 1e-6 | 5 | 0.6 |

Table 9: Training and certification times for STAPS-trained networks.

| Dataset | $\epsilon$ | Train Time (s) | Certify Time (s) |
|---|---|---|---|
| MNIST | 0.1 | 19 865 | 12 943 |
| | 0.3 | 23 613 | 125 768 |
| CIFAR-10 | 2/255 | 47 631 | 398 245 |
| | 8/255 | 48 706 | 77 793 |
| TINYIMAGENET | 1/255 | 861 639 | 35 183 |

test splits and also the best adversarial and certified accuracy (see Figure 7b and Table 12). This indicates that, in this setting, an earlier split and thus larger classifier component induces a (too) difficult optimization problem, leading to worse overall performance.

**Split Position**  We report detailed results for the experiment visualized in Figure 7b (Section 4.2) in Table 11 and Table 12. We additionally report results on TINYIMAGENET in Table 13.

**IBP Regularization**  We repeat the experiment reported on for MNIST in Table 2 for TINYIMAGENET, presenting results in Table 14. We generally observe the same trends, although peak certification performance is achieved slightly later at $\theta_{\text{TAPS}} = 10$ instead of $\theta_{\text{TAPS}} = 5$.

**Repeatability**  Due to the large computational cost of up to 10 GPU-days for some experiments (see Tables 7 and 9), we could not repeat all experiments multiple times to report full statistics. However, we report statistics for the best-performing method for MNIST at $\epsilon = 0.1$ and $\epsilon = 0.3$ and CIFAR-10 at $\epsilon = 2/255$ and $\epsilon = 8/255$ (see Table 15). We generally observe small standard deviations, indicating good repeatability of our results.

## D   Limitations

TAPS and all other certified training methods can only be applied to mathematically well-defined perturbations of the input such as $\ell_p$-balls, while real-world robustness may require significantly more complex perturbation models. Further and similarly to other unsound certified training methods, TAPS introduces a new hyperparameter, the split position, that can be tuned to improve performance further beyond the default choice of 1 ReLU layer in the classifier. Finally, while training with TAPS is similarly computationally expensive as with other recent methods, it is notably more computationally expensive than simple certified training methods such as IBP.

## E   Reproducibility

We publish our code, trained models, and detailed instructions on how to reproduce our results at github.com/eth-sri/taps, providing an anonymized version to the reviewers[3]. Additionally, we provide detailed descriptions of all hyper-parameter choices, data sets, and preprocessing steps in App. B.

---

[3]We provide the codebase with the supplementary material, including instructions on how to download our trained models.

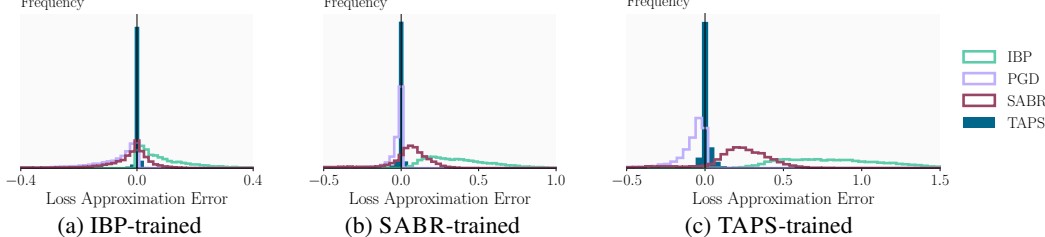

(a) IBP-trained     (b) SABR-trained     (c) TAPS-trained

Figure 9: Distribution of the worst-case loss approximation errors over test set samples, depending on the training and bounding method. Positive values correspond to over-approximations and negative values to under-approximations. We use an exact MILP encoding (Tjeng et al., 2019) as reference.

Table 10: Comparison of TAPS accuracy with certified and adversarial accuracy.

| Dataset | cor(TAPS, cert.) | cor(TAPS, adv.) | TAPS − cert. | TAPS − adv. |
|---|---|---|---|---|
| MNIST | 0.9139 | 0.9633 | $0.0122 \pm 0.0141$ | $0.0033 \pm 0.0079$ |
| CIFAR-10 | 0.9973 | 0.9989 | $0.0028 \pm 0.0095$ | $-0.0040 \pm 0.0077$ |

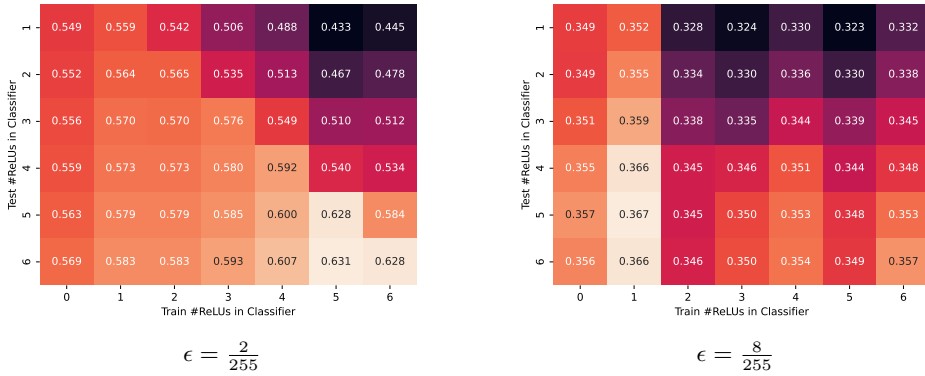

$\epsilon = \frac{2}{255}$        $\epsilon = \frac{8}{255}$

Figure 10: TAPS accuracy of models trained with different classifier sizes for CIFAR-10.

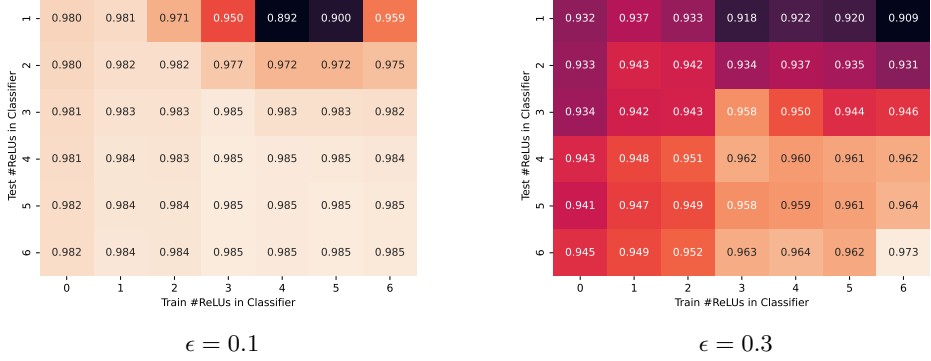

$\epsilon = 0.1$        $\epsilon = 0.3$

Figure 11: TAPS accuracy of models trained with different classifier sizes for MNIST.

Table 11: Effect of split position on accuracies [%] for fixed model size on MNIST.

| $\epsilon$ | # ReLUs in Classifier | Nat. | Adv. | Cert. MN-BaB | Cert. IBP |
|---|---|---|---|---|---|
| | 0 | 98.87 | 98.16 | 98.13 | 97.83 |
| | 1 | 99.06 | 98.37 | 98.31 | 96.27 |
| | 2 | 99.16 | 98.35 | 98.25 | 87.82 |
| 0.1 | 3 | 99.19 | **98.51** | **98.39** | 62.83 |
| | 4 | **99.28** | 98.47 | 98.03 | 4.75 |
| | 5 | 99.22 | **98.51** | 98.17 | 9.76 |
| | 6 | 99.09 | 98.45 | 98.27 | 81.89 |
| | 0 | 97.60 | 93.37 | 93.15 | 93.08 |
| | 1 | 97.94 | 94.01 | **93.62** | 92.76 |
| | 2 | 98.16 | 94.18 | 93.55 | 91.85 |
| 0.3 | 3 | 98.63 | 94.48 | 93.03 | 89.40 |
| | 4 | 98.7 | 94.85 | 93.44 | 89.52 |
| | 5 | 98.63 | 94.64 | 93.26 | 89.15 |
| | 6 | **98.88** | **95.11** | 92.70 | 85.03 |

Table 12: Effect of split position on accuracies [%] for fixed model size on CIFAR-10.

| $\epsilon$ | #ReLUs | Nat. | Adv. | Cert. MN-BaB | Cert. IBP |
|---|---|---|---|---|---|
| | 0 | 67.27 | 56.32 | 56.14 | 53.54 |
| | 1 | 70.10 | 57.78 | 57.48 | 41.86 |
| | 2 | 70.74 | 57.83 | 57.39 | 40.24 |
| $\frac{2}{255}$ | 3 | 71.88 | 58.89 | 58.23 | 34.41 |
| | 4 | 72.45 | 60.38 | 59.47 | 31.88 |
| | 5 | 75.09 | **63.00** | **61.56** | 24.36 |
| | 6 | **75.40** | 62.73 | 61.11 | 24.90 |
| | 0 | 48.15 | 34.63 | 34.60 | 34.26 |
| | 1 | 49.76 | **35.29** | **35.10** | 32.92 |
| | 2 | 47.28 | 33.54 | 33.12 | 28.94 |
| $\frac{8}{255}$ | 3 | 48.76 | 33.50 | 33.12 | 29.14 |
| | 4 | 50.19 | 34.78 | 34.35 | 29.14 |
| | 5 | 50.2 | 34.33 | 33.72 | 28.83 |
| | 6 | **51.03** | 35.25 | 34.44 | 29.97 |

Table 13: Effect of split position on accuracies [%] for fixed model size on TINYIMAGENET.

| ReLU | TAPS Nat. (%) | TAPS Adv. (%) | TAPS Cert. (%) | TAPS Train (s) | TAPS Certify (s) | STAPS Nat. (%) | STAPS Adv. (%) | STAPS Cert. (%) | STAPS Train (s) | STAPS Certify (s) |
|---|---|---|---|---|---|---|---|---|---|---|
| 1 | 28.34 | 20.94 | 20.82 | 306 036 | 23 497 | 28.75 | 22.25 | 22.04 | 350 924 | 35 894 |
| 2 | 27.02 | 20.94 | 20.84 | 944 520 | 32 407 | 28.98 | 22.40 | 22.16 | 861 639 | 35 183 |

Table 14: Effect of IBP regularization and the TAPS gradient expanding coefficient $\alpha$ for TINYIMAGENET $\epsilon = \frac{1}{255}$.

| $w_{\text{TAPS}}$ | Avg time (s) | Nat. (%) | Adv. (%) | Cert. (%) |
|---|---|---|---|---|
| $\mathcal{L}_{\text{IBP}}$ | 0.28 | 25.00 | 19.72 | 19.72 |
| 1 | 1.17 | 25.83 | 20.24 | 20.22 |
| 5 | 2.34 | 28.34 | 20.94 | 20.82 |
| 10 | 4.12 | 28.23 | **21.05** | **20.89** |
| 20 | 5.94 | **28.44** | 20.68 | 20.44 |

Table 15: Mean and standard deviation (over three repeats) for the method with best certified accuracy.

| Dataset | $\epsilon_\infty$ | Method | Nat. [%] | Adv. [%] | Cert. [%] |
|---|---|---|---|---|---|
| MNIST | 0.1 | TAPS | 99.22 ± 0.03 | 98.45 ± 0.06 | 98.28 ± 0.10 |
| | 0.3 | TAPS | 97.96 ± 0.04 | 93.96 ± 0.04 | 93.57 ± 0.02 |
| CIFAR-10 | 2/255 | STAPS | 79.75 ± 0.23 | 65.91 ± 0.12 | 62.72 ± 0.23 |
| | 8/255 | TAPS | 49.07 ± 0.61 | 34.75 ± 0.47 | 34.57 ± 0.46 |

