# OpenReview forum: "Connecting Certified and Adversarial Training"
_NeurIPS.cc/2023/Conference — NeurIPS 2023 poster_

### Official Review · Reviewer_hZus · 2023-07-02

**Soundness:** 4 excellent
**Presentation:** 4 excellent
**Contribution:** 2 fair
**Rating:** 6
**Confidence:** 4

**Summary:**

The paper presents TAPS, an unsound certified training method that combines the advantages of certified training IBP and adversarial training PGD. TAPS first splits the neural network into two parts, the feature extractor, and the classifier. TAPS then uses IBP to propagate the over-approximation through the feature extractor. TAPS uses PGD to estimate multiple adversarial examples inside the over-approximated box and trains with these adversarial examples. The challenge is how to backpropagate gradients through the PGD part in the middle. TAPS designs a gradient estimator to connect the backpropagation. TAPS can also be combined with the current state-of-the-art method, SABR, to reduce the regularization in the feature extractor further, leading to higher natural accuracy and certified accuracy.
The experiment results show that TAPS and STAPS (TAPS+SABR) achieve the highest certified accuracy on MNIST, CIFAR10, and TinyImageNet, except for CIFAR-10 8/255.


**Strengths:**

1. The paper presents a training method that beats the state of the art.
2. The paper conducts extensive abolition studies.

**Weaknesses:**

1. In Table 1, STAPS has the best results in two out of five settings. And TAPS has worse results than SABR in two out of five settings. It seems SABR is as important as TAPS for STAPS. Then the discussion in Section 3.5 needs more rigorous justification. For example, the paper states that "the exponential growth of BOX abstractions still causes a strong regularization of later layers". The experimental illustration of this claim is missing. A layer by layer comparison of Figure 5 would be interesting to see.

**Questions:**

Comment:
1. In line 92, $\bar{\mathbf{o}}^\Delta > 0$ should be  $\bar{\mathbf{o}}^\Delta < 0$.
2. In Section 3.4, between lines 123 and 124, the gradient has an additional factor $2$.

Questions:
1. In lines 162-164, the paper states that the j-th dimension of the latent adversarial examples is independent of the bounds in the i-th dimension. However, PGD will be affected by each dimension. Is this statement in the paper an assumption or these dimensions are in fact independent?
2. In Figure 3, it seems $c$ cannot be greater than 0.5, otherwise the gradient connector cannot be a valid function. However, in Figure 7, $c$ can be larger than 0.5, and even achieves higher natural accuracy. What do I miss here?
3. In Table 2, some settings have a time limit on MN-BaB. Does it means the certified accuracy is over-approximated, e.g., time-outed ones are considered as not certified?

**Limitations:**

The paper does not address any limitation. How to efficiently select hyper-parameters is a large problem for this type of training methods. For example, in Table 2, it might not be possible to completely compute the certified accuracy and to compare settings based on these metrics.

---

> ### Author Rebuttal · Authors · 2023-08-09
>
> We thank Reviewer $\Rf$ for their insightful feedback, helpful suggestions, and interesting questions. Below, we address their questions.
>
> **Q1: Is the independence of a given dimension of the PGD adversarial example from the IBP bounds in a different dimension an assumption?**
> A:  Great question! This depends on the setting on considers:
> When doing an additional step of PGD, the bounds in the jth dimension have no impact on the obtained adversarial example as they impact neither the gradient sign nor the projection in this dimension, as Box bounds are axis parallel. However, when assuming an optimal adversarial attack capable of finding the global minimum over its perturbation space, the resulting adversarial example would indeed depend on all dimension-wise bounds jointly. Thus, this independence holds rigorously (up to initialization) for a single step attack and constitutes a mild assumption otherwise, we are happy to add a corresponding discussion to the relevant section.
>
> **Q2: In Table 2, some settings are noted to have timed out. What does this imply for certified accuracy?**
> A: Typically, branch-and-bound based neural network verifiers such as MN-BaB, the one we use, apply a time-out per sample and consider all samples where verification times out to be uncertified. However, in Table 2, we refer to a total timeout for the verification process. The affected settings in Table 2 are particularly hard to certify, leading to frequent time-outs and thus very long verification times. As the obtained partial results already showed these settings to be less attractive, we decided to only evaluate part of the test set and then report the mean across the evaluated portion. Evaluating the slowest setting would require roughly 100 GPU days.
>
> **Q3: Can you extend the discussion of STAPS in Section 3.5, e.g. expanding on mentioned exponential growth of box abstractions?**
> A: Yes! Please see the main response for a more detailed discussion of STAPS which we are happy to include in Section 3.5.
> While, due to space constraints, Section 3.5 focuses more on outlining the mechanics behind the complementarity of SABR and TAPS rather than a full discussion of all intricacies of STAPS, we are happy to add an extended version of the below discussion and corresponding plots to the appendix.
>
> The exponential growth of Box abstractions, mentioned there, has been established both theoretically and empirically in prior work [1,2] which we are happy to highlight in the relevant section. As Figure 5 shows the worst-case loss approximation error, which can only be computed on the output, a directly analogous layerwise comparison is not possible. While we could compute and compare the mean side lengths of the Box abstractions obtained in different layers, this is very computationally expensive for PGD propagation, as the bound in every dimension requires a separate attack to estimate. However, as the TAPS and IBP and STAPS and SABR bounds are identical in the feature extract (before the split), their pairwise difference shows the isolated effect of the under-approximating effect of the PGD propagation through the classifier. While this can already be seen in Figure 5, we have added versions of the figure showing the distribution of the pairwise differences directly to the PDF attached to the general reply. We are happy to include these in our next revision.
>
> **Q4: Does the gradient in line 213-214 have an extra factor of 2?**
> A: No, when choosing $\alpha = 0.5$, both scaling terms ($2\alpha$ and $(2-2\alpha)$) evaluate to $1$ and we recover the standard gradient as given by the product rule. Choosing different $\alpha$ allows us to scale the gradients of the two loss components in the employed multiplicative regularization, as is common for additive regularizations.
>
> **Q5. Why can the parameter $c$ in the gradient link, illustrated in Figure 3, be greater than 0.5?**
> We believe the confusion might stem from Figure 3 showing two functions corresponding to the partial derivative with respect to the upper and lower bound, respectively. Both can generally have non-zero gradient for the same coordinate of the adversarial example, as is possible for $c > 0.5$
>
> Thanks for pointing out the typo in Line 92, we will correct it.
>
> **References**
> [1] Müller et al. "Certified Training: Small Boxes are All You Need.", ICLR’23
> [2] Shi et al. "Fast certified robust training with short warmup." NeurIPS’21

---

> > ### Comment · Reviewer_hZus · 2023-08-12
> > **Response to Authors**
> >
> > Thanks for authors' efforts and detailed replies. I will raise my score. Overall, I'd like to see this paper being accepted.
> >
> > For Q1, please add a corresponding discussion to the relevant section. For Q2 please add how do you compute the numbers, i.e., "evaluate part of the test set and then report the mean across the evaluated portion", to the paper or appendix.

---

### Official Review · Reviewer_5pzA · 2023-07-06

**Soundness:** 2 fair
**Presentation:** 3 good
**Contribution:** 2 fair
**Rating:** 5
**Confidence:** 4

**Summary:**

This paper aims to improve the combination of adversarial training and certified training for certified robustness. A gradient connector is proposed for jointly conducting these two kinds of training. Results show some improvement on the certified robustness after the training, as well as a more precise approximation on the worst-case loss.

**Strengths:**

- This paper proposes a "gradient connector" which enables end-to-end training with both adversarial training and IBP-based certified training.
- There is some empirical improvement on the certified accuracy on some settings (TinyImageNet), compared to prior works combining adversarial training and IBP training.
- The proposed method can more precisely estimate the worst-case loss, compared to PGD, iBP, or prior methods doing the combination.

**Weaknesses:**

The empirical improvement on the major metric, certified accuracy, is very marginal and sometimes negative:
- Compared to SABR (Muller et al., 2022a), the absolute improvement on MNIST is only 0.17% or 0.22%.
- On CIFAR, simply using the proposed method does not bring any improvement but may even yield lower certified accuracy.
- On CIFAR eps=2/255, combining SABR and the proposed method only has an absolute improvement of 0.14%. On CIFAR eps=8/255, doing this combination still leads to worse results compared to SABR alone.

Overall, the current results on all the three datasets do not sufficiently demonstrate that the proposed method is effective in practice. It is still possible that the tiny difference on MNIST and CIFAR may come from randomness, yet standard deviation is not reported.

==Updates==

Thanks to the authors for the explanations. I understand that both TAPS and STAPS are contributions of this work. However, while this paper looks overall good, empirical results still look kind of weak to me (as mentioned in the first point and the third point in my original review). Thus I am maintaining my original rating.

**Questions:**

See "Weaknesses".

**Limitations:**

Not discussed.

---

> ### Author Rebuttal · Authors · 2023-08-09
>
> We thank Reviewer $\Rtr$ for their insightful feedback, helpful suggestions, and interesting questions.  We are delighted they appreciate our novel gradient connector and found our performance improvements on the challenging TinyImageNet satisfactory. Before addressing the reviewer’s remaining questions, we would like to respectfully ask them to outline any soundness concerns they might have (leading to a soundness score of 2) as their review does not touch on any such point.
>
> **Q1: Can you comment on the magnitude of improvement of TAPS and STAPS over SABR?**
> A: First, we want to highlight, that both TAPS and STAPS are core contributions of this work and thus believe the fact that TAPS alone can realizes a comparable or bigger improvement over prior work than SABR (which achieved the biggest improvement in years) to highlight the promise of our method rather than being a weakness.
> Second, in settings where low regularization strength is particularly desirable (CIFAR-10 2/255 and TinyImageNet 1/255), TAPS and SABR can be efficiently combined to STAPS to achieve even higher performance and highlighting the complementarity of the two methods (discussed in more detail in Section 3.5).
> Third, while the absolute improvements on MNIST might be small, they are a much bigger than e.g. those of SABR over SortNet and correspond to significant portions of the remaining error (9.5% and 3.3%), highlighting their importance.
>
> **Q2: Can you report the standard deviation of the considered performance metrics**
> We first want to highlight that we already report standard deviations for MNIST in Table 14. We are happy to also report statistics for all other settings in the next revision of this work. However, given that prior work often only reports the best observed results, putting these results in the right context would require reproducing and reporting statistics for all baseline methods as well, thus carrying significant computational costs.

---

### Official Review · Reviewer_8yjL · 2023-07-06

**Soundness:** 4 excellent
**Presentation:** 3 good
**Contribution:** 3 good
**Rating:** 7
**Confidence:** 3

**Summary:**

This paper proposes a method called Training via Adversarial Propagation through Subnetworks (TAPS) for improving certified adversarial robustness. Specifically, TAPS splits the network $f$ into a feature extractor $f_E$ and a classifier $f_C$. During training, TAPS first uses interval bound propagation (IBP) to bound the feature extractor's exact reachable set in the embedding space and conducts adversarial attacks (PGD) in the embedding space to the classifier. This method can also be combined with other state-of-the-art methods like SABR (called STAPS) that can further improve its performance.

## post-rebuttal
I've updated my score since my concerns are adequately addressed.

**Strengths:**

1. This paper is well-motivated and well-organized.
2. The idea of connecting adversarial training and IBP training is novel and impressive.
3. The experiments show that the proposed method can outperform state-of-the-art methods for several settings.

**Weaknesses:**

1. The certified robustness for a larger perturbation bound (8/255) is not comparable with SortNet, showing the limitation of TAPS/STAPS under certain settings. Additionally, the comparison under a larger perturbation bound (>1/255) for TinyImageNet is not provided. Based on the comparison under 8/255 for CIFAR-10, this reviewer infers that SortNet may also outperform TAPS/STARPS for a larger dataset with a larger perturbation bound.
2. The details of how to combine TAPS and SABR are not provided in Section 3.5.
3. This work only focuses on $\ell_\infty$ robustness and does not show scalability to other metrics of robustness, such as $\ell_2$-norm.
4. It seems that several experiments in this paper were not completed when submitted (Tables 2 and 3). However, I think this is acceptable since the main comparison is complete. The authors should complement the missing results if accepted.

### Minor Comments

1. Line 11. I personally suggest removing the claim of publication for your implementation and networks, even though you have anonymized the code link. As this paper is still under review, the code has not yet been published.
2. Line 44. Lack of the full spelling and reference for SABR. Also, the details for SABR are not sufficiently introduced in this paper.
3. Line 105. I suggest replacing the claim "vulnerable" of adversarial training under stronger attacks with a more moderate tone. Even under AutoAttack, the robustness of an adversarially trained model is slightly lower than PGD. So far, adversarial training is still one of the most effective methods to improve adversarial robustness. Thus, this assertion on adversarial training methods is unfair for this research area and may mislead the community.
4. Line 148. Is $\theta_F$ exactly $\theta_E$?
5. Line 595. The author stated that they provided detailed descriptions, but the location is shown as "??".

**Questions:**

1. Line 36: Can you explain what "tractable" means in this context?
2. Lines 114-115: How is $x'$ selected for SABR? I suggest adding more details here since SABR is the main baseline of your method.
3. Although the code is provided, this reviewer strived to understand it. In particular, where is the command and variable for splitting the network?
4. What is the constraint ($\epsilon$-ball) for PGD attack in TAPS? Since PGD is conducted in the embedding space, the constraint on the input space may not be effective.
5. How about using PGD for the feature extractor and IBP for the classifier?

---

> ### Author Rebuttal · Authors · 2023-08-09
>
> We thank Reviewer $\Rt$ for their insightful feedback, helpful suggestions, and interesting questions. Below, we address their questions.
>
> **Q1: Why is only $\ell_\infty$-norm robustness and no $\epsilon$ larger than $1/255$ for TinyImageNet evaluated? How would SortNet perform in such a setting?**
> We follow the conventions of the field in both of these respects: First, works investigating convex relaxation based certified robustness (such as TAPS) almost always investigate only $\ell_\infty$-norm robustness [1,2,3,4,5,...]. Second, obtaining non-trivial certified accuracy on TinyImageNet is already highly challenging for $\epsilon = 1/255$, thus larger radii are typically not considered in the literature [1,3,4,6] (we are not aware of any results). Further, while SortNet’s [6] performance on CIFAR-10 for $\epsilon = 8/255$ is impressive, we believe this to not necessarily be an indication of great performance on TinyImageNet at larger radii. In fact, on TIN at $\epsilon = 1/255$ it is only on par with IBP and on MNIST, it is dominated by STAPS regardless of radius.
>
> **Q2: Can you expand the background on SABR and add how TAPS and SABR combine to STAPS?**
> Please see the main response for a detailed reply!
>
> **Q3: Are the results in Tables 2 & 3 complete?**
> A: Great question, we will clarify this in the text. Table 3 is complete and missing results simply show the instability of single-estimator PGD, thus highlighting the importance of our multi-estimator PGD.
> In Table 2, we decided to not complete the evaluation of the last two rows as this would require well over 100 GPU days and the partial results already show a severe drop in certified accuracy.
>
> **Q4: Can you clarify the meaning of ``tractable'’ (L36)?**
> A: Generally the verification problem is NP-hard [7]. However, recent branch-and-bound-based approaches [8] can solve many practical instances efficiently. We will clarify this.
>
> **Q5: In your code, what is the command and parameter for splitting the network?**
> A: TAPS is implemented in ```torch_model_wrapper.py``` in the class ```BoxModelWrapper```. The method ```split_net_to_blocks``` of this class splits the network into feature extractor and classifier. When the code is publicly released, we will provide documentation containing such details.
>
> **Q6: What are the constraints/bounds for PGD in the embedding space utilized in TAPS?**
> A: As we outline in L133ff, we first propagate $\mathcal{B}(x, \epsilon)$ via IBP through $f_E$. This results in interval bounds $[\underline{\mathbf{z}}, \overline{\mathbf{z}}]$, describing a hyper-rectangle (i.e., a stretched $\ell_\infty$-ball). We then conduct an adversarial attack using PGD within this hyper-rectangle in the latent space of the model.
>
> **Q7: Could you use PGD for the feature extractor and IBP for the classifier?**
> A: While this is possible in theory, in practice this is infeasible. For each bound (upper and lower) in each dimension we require 1 PGD attack. As the latent space in many layers has over 20.000 dimensions, this requires large amounts of compute and memory per sample. Note, that when using the last layer, we only need to upper bound logit differences and thus require only (#classes - 1) attacks.
>
>
> **References**
> [1] Müller et al. "Certified Training: Small Boxes are All You Need.", ICLR’23
> [2] Balunovic and Vechev. "Adversarial training and provable defenses: Bridging the gap." ICLR’19
> [3] De Palma et al. "IBP regularization for verified adversarial robustness via branch-and-bound."
> [4] Shi et al. "Fast certified robust training with short warmup." NeurIPS’21
> [5] Zhang et al. “Towards Stable and Efficient Training of Verifiably Robust Neural Networks” ICLR’20
> [6] Zhang et al. ““Rethinking lipschitz neural networks and certified robustness: A boolean function perspective” arXiv
> [7] Katz et al. "Reluplex: An efficient SMT solver for verifying deep neural networks." CAV’17
> [8] De Palma et al. "Improved branch and bound for neural network verification via lagrangian decomposition." arXiv

---

> > ### Comment · Reviewer_8yjL · 2023-08-10
> >
> > Dear authors,
> >
> > Thanks for the rebuttal. Most of my concerns have been addressed, and I will raise my score when review editing is allowed.
> >
> > For Q4, I am still a bit confused about what is ``tractable``. Is this means the verification problem is not NP-hard?
> >
> > Additionally, please repaint my name with red color, which is my favorite color.
> >
> > Best,

---

> > > ### Author Response · Authors · 2023-08-10
> > > **On the tractability of neural network verification**
> > >
> > > We thank Reviewer $\textcolor{red}{8yjL}$ for their quick reply and are happy that we were able to address all their concerns.
> > >
> > > **Tractability of the Neural Network Verification Problem**
> > >
> > > Generally, neural network verification remains NP-hard. However, as with many NP-Hard problems, many instances are efficiently decidable (think SAT/SMT solvers). In the case of neural network verification, wether an instance is is efficiently solvable depends on the combination of input, robustness specification, network, and verifier.
> > > Until recently, so called incomplete verification methods were commonly used, which (typically) have a fixed precision and can either decide a property or return that the result is unknown. Such verifiers (like IBP) were generally only able to verify networks heavily regularised towards verifiability (at the cost of significantly reduced standard accuracy).
> > > While complete verification methods can decide any neural network verification property given sufficient (in the worst case exponential) time, these methods were typically based on mixed integer linear programming or SAT/SMT solvers and to inefficient for neural networks of relevant size. However, recently much more efficient Branch-and-Bound based complete verifiers have been proposed which are efficient enough to be applied to much less heavily regularized networks. This is what we mean by their certification has become (practically) tractable.
> > >
> > > $\textcolor{red}{\text{Unfortunately, we can not change the color of reviewer }} \textcolor{red}{8yjL \text{ in the main response, however, we hope they enjoy this red text.}}$

---

> > > > ### Comment · Reviewer_8yjL · 2023-08-12
> > > >
> > > > Dear authors,
> > > >
> > > > Thanks for the further reply. I've updated my rating. Hope this paper can make a great contribution into this research area.
> > > >
> > > > Best,

---

### Official Review · Reviewer_jSFD · 2023-07-07

**Soundness:** 3 good
**Presentation:** 3 good
**Contribution:** 3 good
**Rating:** 7
**Confidence:** 5

**Summary:**

The paper proposes TAPS -- a method to combine IBP and PGD to train better certified networks. The authors observe that IBP on its own leads to overestimation of the inner adversarial loss, where as PGD leads to an underestimation. Therefore, intuitively, the approximation errors may compensate for each other during training. Thus, TAPS leverages PGD on the pre-classification logits output using IBP to generate a better loss approximate. Given the non-differentiable nature of this, the authors propose a rectified linear gradient approximation to allow end to end training. Empirical results are provided for CIFAR10, MNIST and TinyImagenet, showing improvement over pure IBP and other IBP-approximation approaches.

**Strengths:**

1. The proposed approach is well-motivated and the authors provide clear justification through experimental analysis.
2. The gradient connector is novel mechanism allowing end to end training of PGD + IBP networks. This is a creative approach towards combining two connected but complementary methods, essentially improving on COLT which sequentially leverages both.
3. The experiments are thorough and support the claims in the paper.
4. The paper is well-written and thoughtfully explains the intuition for every step in the algorithm.

**Weaknesses:**

While not a major weakness, I would be interested to see if leveraging stronger or weaker attacks during the PGD training step significantly changes results. Perhaps a simple experiment with varying PGD iterations, or even a stronger attack like Autoattack would clearly answer this.

**Questions:**

1. What is the computational overhead of TAPS/STAPS over other methods?

**Limitations:**

The authors have clearly mentioned limitations.

---

> ### Author Rebuttal · Authors · 2023-08-09
>
> We thank Reviewer $\Ro$ for their insightful feedback, helpful suggestions, and interesting questions. Below, we address their questions.
>
> **Q1. What is the computational overhead of TAPS/STAPS over other methods?**
> When using single-estimator PGD, TAPS is strictly faster than SABR [1], as it only requires a partial IBP propagation and adversarial search only over the classifier component. When using multi-estimator PGD, the runtime trade-off depends on the number of classes and size of the classifier component. Here, TAPS requires multiple PGD attacks over a smaller network (component) while SABR requires a single attack over the whole network. Compared to TAPS, STAPS requires an additional adversarial attack over the whole network and is thus always slightly slower than SABR. However, both TAPS and STAPS are notably faster than COLT [2], as IBP propagation is much faster than DeepZ [3] and the gradient connector makes COLT’s complex training in multiple stages obsolete. For TinyImageNet, we find TAPS to already be consistently slower than SABR but believe that designing strategies between single- and multi-estimator PGD to be an interesting item for future work that has the potential to change this. We provide runtimes in Table 6 and 8 in Appendix B, for TAPS and STAPS, respectively.
>
> **Q2: What is the effect of the adversarial attack’s strength on the obtained results?**
> Great questions! We have conducted a corresponding experiment using 1 to 100 attack steps with 1 or 3 restarts to investigate this for MNIST $\epsilon=0.3$ and the CNN7 architecture used for our main results and report results in the table below. Interestingly, even a single attack step and restart are sufficient to achieve good performance with TAPS, in particular, outperforming standard IBP. As we increase the strength of the attack, we can increase certified accuracy slightly while marginally reducing natural accuracy, agreeing well with our expectation of regularization strength increasing with attack strength. We are happy to include these results in the next revision of the paper. We also provide nicely rendered version of this table in the PDF attached to the general reply.
>
>
> | Restarts | Number of Steps |   Natural  |  Certified |
> |:--------:|:---------------:|:----------:|:----------:|
> |     1    |        1        | **0.9822** |   0.9336   |
> |          |        5        |   0.9790   |   0.9315   |
> |          |        20       |   0.9778   |   0.9343   |
> |          |       100       |   0.9794   | **0.9346** |
> |     3    |        1        | **0.9822** |   0.9347   |
> |          |        5        |   0.9790   | **0.9355** |
> |          |        20       |   0.9799   |   0.9352   |
> |          |       100       |   0.9799   | **0.9355** |
>
> **References**
> [1] Müller et al. "Certified Training: Small Boxes are All You Need.", ICLR’23
> [2] Balunovic and Vechev. "Adversarial training and provable defenses: Bridging the gap." ICLR’19
> [3] Singh et al. "Fast and effective robustness certification." NeurIPS’18

---

> > ### Comment · Reviewer_jSFD · 2023-08-14
> > **Comment on the rebuttal**
> >
> > Thank you for answering my questions. After reading through the other reviews and the rebuttal, I maintain my earlier score. Overall, I find the paper to present an interesting approach to connecting adversarial training and certification based methods.

---

### Author Rebuttal · Authors · 2023-08-09

$\newcommand{Ro}{\textcolor{purple}{jSFD}}$
$\newcommand{Rt}{\textcolor{green}{8yjL}}$
$\newcommand{Rtr}{\textcolor{blue}{5pzA}}$
$\newcommand{Rf}{\textcolor{orange}{hZus}}$


We thank all reviewers for their insightful feedback, helpful suggestions, and interesting questions. We were encouraged that they found our work well-motivated ($\Ro$, $\Rt$), novel ($\Ro$, $\Rt$) and well supported by our state-of-the-art empirical results ($\Ro$, $\Rt$, $\Rtr$ $\Rf$) and extensive ablations ($\Rf$). We answer the sole shared question here, before addressing the reviewer-specific ones in individual responses and look forward to the reviewers’ replies.

**Q1: Can you expand the discussion on STAPS in Section 3.5 and the relevant background on SABR? ($\Rt$, $\Rf$)**
Yes! We are happy to extend the background on SABR as well as the discussion of STAPS in Section 3.5, as outlined below.

At a high level: IBP training propagates the entire input region $\mathcal{B}(x, \epsilon)$ for each sample $x$ in order to evaluate and then optimize the IBP loss (Eq. (2)).
In contrast, SABR propagates only a small subset $\mathcal{B}(x’, \tau) \subseteq \mathcal{B}(x, \epsilon)$, chosen by performing an adversarial attack to select $x’$, but otherwise uses the same loss and training procedure. In STAPS, we simply replace the IBP component with SABR, thus only propagating a small subset $\mathcal{B}(x’, \tau) of the input region, selected using an adversarial attack, to compute the bounds for the adversarial attack in the latent space of the feature extractor.

**In the attached PDF we provide a version of Figure 5 that highlights the difference between IBP and TAPS (as well as SABR and STAPS) for Q3 of $\Rf$.**

---

### Decision · Program_Chairs · 2023-09-21

**Decision:**

Accept (poster)

**Comment:**

I have read all the submitted reviews, rebuttals, and ensuing discussions. The initial reviews were generally positive, and I'm pleased to note that the authors' rebuttal has effectively addressed the concerns raised by some reviewers. All reviewers recognize the contributions and novelty of the work, concurring that this is indeed a high-quality contribution.

The primary concern emerged from two reviewers, highlighting the somewhat less competitive results obtained on MNIST and in one particular setup of CIFAR. A discussion to address these points would be beneficial. Furthermore, the overall discussion of STAPS would enhance the paper's readability.

I would also suggest that the authors consider citing [A]. As far as I understand, [A] is the first approach to marrying certified and empirical methods within the training framework to enhance certified accuracy. Notably, they undertake adversarial training (PGD) on the smooth certified classifier using randomized smoothing. While closely related to this work, the distinction lies in the certification approach, where this paper employs IBP.

In summation, my recommendation is to accept the paper.

[A] Provably Robust Deep Learning via Adversarially Trained Smoothed Classifiers